# Evaluation of Neuropsychiatric Effects of Montelukast–Levocetirizine Combination Therapy in Children with Asthma and Allergic Rhinitis

**DOI:** 10.3390/children10081301

**Published:** 2023-07-28

**Authors:** Uğur Altaş, Zeynep Meva Altaş, Fırat Öz, Mehmet Yaşar Özkars

**Affiliations:** 1Department of Pediatric Allergy and Immunology, University of Health Sciences, Ümraniye Training and Research Hospital, Ümraniye, 34764 Istanbul, Türkiye; myozkars@hotmail.com; 2Ümraniye District Health Directorate, Ümraniye, 34764 Istanbul, Türkiye; zeynep.meva@hotmail.com; 3Department of Child and Adolescent Psychiatry, Siirt Training and Research Hospital, 56000 Siirt, Türkiye; firatoz_92@hotmail.com

**Keywords:** asthma, allergic rhinitis, children, neuropsychiatric effects, montelukast, levocetirizine, combination therapy

## Abstract

Drug-induced neuropsychiatric effects are important for disease management. We aim to evaluate the neuropsychiatric effects of montelukast–levocetirizine combination therapy in children. This descriptive study was conducted with children aged 2–5 years, diagnosed with asthma and allergic rhinitis, who began to receive montelukast and levocetirizine combination therapy. The respiratory and asthma control test for children (TRACK), Rhino Conjunctivitis Scoring System (RCSS), and common neuropsychiatric effects (irritable behavior, hallucinations, headaches, nightmares, sleep disorders, behavioral and mood disorder, restlessness, depression) were ascertained by the questionnaire applied before and 4 weeks after the treatment. Parents answered on behalf of their children. The most common finding before and after treatment was irritable behavior. While irritable behavior was observed in 82.4% (*n* = 56) of children before the treatment, this percentage was 63.2% (*n* = 43) after the treatment (*p* = 0.004). The percentage of children who developed at least one neuropsychiatric symptom after treatment was 22.1% (*n* = 15). There was no significant effect of age, gender, RCSS, TRACK, or allergy test positivity on the development of neuropsychiatric symptoms (*p* > 0.05). According to the results, at least one neuropsychiatric finding developed in approximately one in five children. Identifying risk factors will enable more careful treatment or consideration of alternative treatments for children at higher risk in the clinical follow-up period.

## 1. Introduction

The prevalence of allergic diseases is increasing worldwide and allergic diseases are an important public health issue [1,2]. Asthma is one of the most common allergic and chronic diseases of childhood, characterized by airway hypersensitivity and chronic respiratory tract inflammation, and presents with recurrent attacks of wheezing, cough, shortness of breath, and chest pain [3]. These asthma-related symptoms usually begin in the early age periods (pre-school period, before 5 years of age) [4]. The Centers for Disease Control and Prevention (CDC) report that more than 4 million children have a diagnosis of asthma [5].

Currently, there is no method of treatment that can prevent the development of asthma or change its natural course in long-term follow-ups. However, the disease can be controlled with treatments applied in clinical practice for asthma. The general goal in the treatment of asthma is to reduce the symptoms associated with the disease and to keep the disease under control. Response to treatment should be evaluated in patients approximately 2–3 months after the start of treatment for asthma [6,7]. During control visits, the patient should be questioned in terms of asthma-related findings, and treatment-related conditions, such as improvement in symptoms and drug-related side effects, should be evaluated in detail [6].

Allergic rhinitis is one of the most common allergic diseases along with asthma, and they can frequently coexist in a patient [8]. Immunoglobulin E-mediated nasal mucosal inflammation against allergens is the underlying mechanism in the pathophysiology of allergic rhinitis [9]. Common clinical manifestations of allergic rhinitis include sneezing, itching and runny nose, nasal congestion, and redness and tearing of the eyes [10]. The causes of these symptoms are underlying inflammatory process and/or a dysfunctional nasal mucosa. Allergic rhinitis causes a significant burden of disease worldwide [11]. For this reason, diagnosis, effective treatment, and follow-up period of allergic rhinitis is extremely important for the health of the patients.

The initiation of appropriate treatment after diagnosis and monitoring of response to treatment in chronic diseases are extremely important in disease control. Low-dose inhaled corticosteroids are the first-line initial treatment for asthma control in children aged 5 years and younger. It is recommended to start an inhaled short-acting beta 2 agonist to reduce symptoms in all children with asthma attacks [7]. Montelukast is also used in the treatment and prophylaxis of asthma attacks [12]. As an alternative or combination therapy to or with oral antihistamines or nasal corticosteroids, montelukast can be given to patients with allergic rhinitis [13].

Montelukast, a leukotriene receptor antagonist drug, was licensed by the American Food and Drug Association (FDA) in 1998 [14]. Leukotrienes are arachidonic acid metabolites produced by the action of the 5-LO (5-lipoxygenase) enzyme [15]. There are data showing that leukotrienes are synthesized in the lung following antigen provocation and are elevated in asthma [16]. Leukotrienes mediate the pathophysiology of diseases accompanied by inflammation, such as asthma and allergic rhinitis, by binding to the receptors of leukotrienes in the upper and lower respiratory tracts [17]. Regarding asthma, leukotrienes mediate bronchoconstriction [18]. In allergic rhinitis, leukotrienes that bind to nasal receptors increase vascular permeability, causing nasal congestion, increased mucus production, and secretion [17].

Montelukast has an important role in disease and symptom control in asthma and allergic rhinitis. However, there are studies have reported that montelukast may cause some neuropsychiatric side effects [19,20]. Although side effects associated with the use of montelukast are rare and usually mild, they can be worrisome for patients and their families and may negatively affect their quality of life [21]. There are assumptions that the neuropsychiatric effects of the drug develop when it crosses the blood–brain barrier and blocks leukotriene receptors in the brain [22]. The common neuropsychiatric effects of montelukast include irritable behavior, agitation, restlessness, nightmares, depression, anxiety, hallucinations, sleep-related disorders, and suicidal ideation and attempt [23,24,25,26]. In 2020, the FDA declared the need for a Boxed Warning for montelukast due to serious mental side effects such as suicidal ideation. It is recommended that it be prescribed by considering the benefit–harm relationship for asthma patients, and its use should not be preferred in the presence of other treatment alternatives for allergic rhinitis patients [27].

Levocetirizine is a second-generation antihistamine and is frequently used to control the symptoms of allergic rhinitis. It acts by blocking H1-histamine receptors. Side effects related to levocetirizine are less common than other second-generation antihistamine drugs [28].

In a study evaluating the neuropsychiatric side effects of drugs that can be used in the treatment of asthma and allergic rhinitis, the most adverse effects were seen in those using montelukast [29]. In the same study, neuropsychiatric side effects were observed more frequently in patients using levocetirizine, one of the H1-antihistamine drugs, compared to patients using inhaled corticosteroids. When the literature was reviewed, no study was found that evaluated the use of montelukast and levocetirizine in combination in terms of the development of neuropsychiatric effects.

Knowing the drug-induced neuropsychiatric effect profile in patients is important for continuity of treatment and disease management. In our study carried out in this context, our aim was to evaluate the neuropsychiatric effects of montelukast and levocetirizine combination therapy in children aged 2–5 years with asthma and allergic rhinitis.

## 2. Materials and Methods

### 2.1. Study Design, Type, and Sample

The descriptive study was conducted with children aged 2–5 years who were diagnosed both with asthma and allergic rhinitis, who applied to our pediatric allergy and immunology outpatient clinic of a tertiary hospital located in Istanbul. Children older than 5 years of age or younger than 2 years of age, patients with a previous diagnosis of neuropsychiatric disease, and patients who were previously treated with montelukast and/or levocetirizine were excluded from the study. Patients who had allergic rhinitis and asthma together after their outpatient clinic examination and who were started on the combination therapy of montelukast and levocetirizine for the first time were included in the study. A combined preparation containing 4 mg montelukast and 2.5 mg levocetirizine was used in the treatment for all patients in our study. During the approximately 3-month data collection period, the study was conducted with those who met the inclusion criteria and gave consent to participate. Parental consent was obtained on behalf of the children. All data were obtained via a questionnaire applied face-to-face.

### 2.2. Evaluations

The same type of questionnaire was applied to the patients before and 4 weeks after the beginning of the treatment. Parents answered the questionnaire on behalf of their children. The respiratory and asthma control test for children, the Rhino Conjunctivitis Scoring System (RCSS), and the presence of 7 common neuropsychiatric effects (irritable behavior, hallucinations, headaches, nightmares, sleep disorders, behavioral and mood disorder, restlessness, depression) were included in the questionnaire. When questioning the presence of each neuropsychiatric finding, the answers were 4-point Likert type: none (0), rarely (1), often (2), and always (3). The development of neuropsychiatric findings and improvement in neuropsychiatric findings were determined according to the change in Likert-type responses performed before and after the treatment. For example, if a patient was rarely restless before the treatment, and had restlessness frequently or always after the treatment, it was interpreted that this patient developed restlessness as a neuropsychiatric finding. Similarly, if a patient responded always experiencing hallucinations before the treatment, and this response changed to frequently, rarely, or none after the treatment, it was interpreted that the hallucination finding improved in this patient. In addition, the sociodemographic characteristics of the patients, such as age, gender, and allergy tests; and clinical and laboratory characteristics, such as total IgE, WBC, neutrophil, eosinophil, lymphocyte, and platelet values were evaluated in the study.

#### 2.2.1. Test for Respiratory and Asthma Control in Kids (TRACK)

The TRACK questionnaire was developed for use in children under the age of five [30]. The questionnaire consists of five different questions: frequency of symptoms such as wheezing, cough, and shortness of breath in the last four weeks, frequency of night awakenings due to symptoms, limitation of activity due to symptoms, frequency of use of bronchodilator drugs in the last three months, and frequency of oral corticosteroid use in the last year. The questionnaire is a 5-point Likert scale and each question is scored between 0, 5, 10, 15, and 20. The total score ranges from 0–100, with higher scores indicating better disease control. A score of 80 or more is considered well-controlled asthma [30,31].

#### 2.2.2. Rhino Conjunctivitis Scoring System (RCSS)

The RCSS was used for investigation of the severity of AR symptoms. Six symptoms, including nasal itching, nasal congestion, rhinorrhea, sneezing, redness of the eye, and watery eyes, are evaluated with the RCSS. Each symptom is scored by patients as 0 (none), 1 (mild), 2 (moderate), or 3 (severe). The sum of the scores for each six symptoms is divided into six for the calculation of the total RCSS [32].

### 2.3. Statistical Analysis

The SPSS (Statistical Package for Social Sciences) for Windows 25.0 program was used for data analysis and recording. The descriptive data were presented as median, minimum, maximum values, numbers (n), and percentages (%). Conformity of continuous variables to normal distribution was visual (histogram and probability graphs) and analytical methods (Kolmogorov–Smirnov/Shapiro–Wilk tests). The Mann–Whitney U test was used to compare non-normally distributed continuous variables. The McNemar test was used for comparison of two related groups (percentage of neuropsychiatric findings seen before and after treatment). Logistic regression modeling was performed as a multivariate analysis to evaluate the factors associated with the development of neuropsychiatric findings. *p* < 0.05 was considered as the statistical significance level.

### 2.4. Ethics

Ethics committee approval was obtained from the University of Health Sciences, Ümraniye Training and Research Hospital Ethics Committee on 22/12/2022 with decision number 417.

## 3. Results

In the study, 68 children with a diagnosis of allergic rhinitis and asthma who were treated with a montelukast and levocetirizine fix combination therapy were evaluated. While 57.4% (*n* = 39) of the children were boys, 42.6% (*n* = 29) were girls. The median age was 3 years (2.0–5.0).

When the laboratory values of the patients were examined, the median WBC and neutrophil values were 8480.0 103/uL (3330.0–21,440.0) and 3450.0 103/uL (1080.0–14,500.0), respectively. Absolute eosinophil and eosinophil (%) values were 260.0 103/µL (0–1760.0) and 3.0% (0–16.9). The lymphocyte, platelet, and total IgE values of the patients were 3840.0 103/uL (1140.0–13,510.0), 334,000.0 103/uL (167,000.0–743,000.0), and 91.0 IU/mL (2.0–2735.0) (Table 1).

When the allergy test results of the patients were examined, 33.8% (*n* = 23) had a positive allergy test against at least one allergen. Of the children 20.6% (*n* = 14) had a house dust mite allergy, 7.4% (*n* = 5) had an egg allergy, and 4.4% (*n* = 3) had a cat allergy. While the rate of children with a nut allergy was 4.4% (*n* = 3), the number of children with a pollen or cow’s milk allergy was 2 (2.9%) for both (Table 2).

TRACK scores, which are used to evaluate the asthma control of the patients, were evaluated before and after the treatment in the study. According to the TRACK questionnaire, children’s clinical signs of asthma improved significantly after treatment. The median value of TRACK score before treatment [40.0 (5.0–75.0)] significantly increased after treatment [87.5 (25.0–100.0)] (*p* < 0.001). Similarly, the RCSS score, which is used to evaluate clinical severity over the findings of allergic rhinitis, was evaluated before and after treatment. After the treatment, a significant improvement was observed in the clinical findings of allergic rhinitis. The median RCSS score before treatment significantly decreased after treatment [2.17 (0.83–3.0) and 0 (0–3.0), respectively] (*p* < 0.001) (Table 3).

Common neuropsychiatric findings of the children before and after the treatment were evaluated. There was no child with depression before and after treatment. All findings showed a decrease in frequency after treatment compared to before. This decrease was statistically significant, excluding hallucinations. The most common finding before and after treatment was irritable behavior. While irritable behavior was observed in 82.4% (*n* = 56) of the children before the treatment, this percentage was 63.2% (*n* = 43) after the treatment (*p* = 0.004). The frequency of hallucinations was 11.8% (*n* = 8) and 7.4% (*n* = 5) before and after treatment (*p* = 0.508). Headache was seen in 20.6% of the children (*n* = 14) before the treatment, while only 2 children had headaches after the treatment (*p* < 0.001). The frequency of nightmares was 51.5% (*n* = 35) and 25.0% (*n* = 17) before and after treatment, respectively (*p* < 0.001). Sleep disorder was observed in 63.2% (*n* = 43) of the children before treatment and in 32.4% (*n* = 22) after treatment (*p* < 0.001). The frequency of behavioral and mood disorders was 36.8% (*n* = 25) and 22.1% (*n* = 15) before and after treatment, respectively (*p* = 0.013). While restlessness was seen in 45.6% (*n* = 31) of the children before treatment, it was seen in 23.5% (*n* = 16) after treatment (*p* < 0.001) (Table 4).

The percentage of children who developed at least one neuropsychiatric symptom after treatment was 22.1% (*n* = 15). The most common finding was irritable behavior (*n* = 9, 13.2%). Headache and depression were not observed in any child. The frequency of nightmares and sleep disorders were 7.4% (*n* = 5) and 4.4% (*n* = 3), respectively. Behavior and mood disorders were seen in 5.9% (*n* = 4) of the children. Hallucinations were seen in 4.4% (*n* = 3) of the children, while restlessness was seen in 2 (2.9%) children (Table 5).

The percentage of improvement in at least one neuropsychiatric symptom was 76.5% (*n* = 52). The most common improvements were observed in sleep disorder (45.6%, *n* = 31), irritable behavior (36.8%, *n* = 25), and nightmares (35.3%, *n* = 24), respectively. The percentage of children with improvement in restlessness was 27.9% (*n* = 19). The percentage of children with improvement in behavior and mood disorders was 26.5% (*n* = 18). Improvement was observed for headache in 17.6% (*n* = 12) and for hallucinations in 10.3% (*n* = 7) of the children (Table 5).

Logistic regression analysis was used to evaluate the factors that may be associated with the development of at least one neuropsychiatric finding. While the dependent variable in the analysis was the development of neuropsychiatric findings, the independent variables were gender, age, TRACK (before treatment), RCSS (before treatment), and allergy test positivity. There was no statistically significant effect of any of the independent variables on the development of neuropsychiatric symptoms (*p* > 0.05) (Table 6).

## 4. Discussion

In this study, which was conducted to evaluate the neuropsychiatric effects of montelukast and levocetirizine combination therapy, which are frequently used in the treatment of asthma and allergic rhinitis [12,33], some neuropsychiatric findings that can be seen frequently before and after treatment were questioned. The percentage of children who developed at least one neuropsychiatric symptom after treatment was 22.1%. In a study conducted in pediatric patients in our country, 62.4% of children had side effects after treatment with montelukast [23]. In a different study in the literature, the percentage of development of neuropsychiatric symptoms after the use of montelukast in children with asthma was reported as 32% [34]. In our study, the percentage of development of neuropsychiatric findings was found to be lower than similar studies in the literature. This may be related to the combined use of montelukast and levocetirizine in our study, and the use of only montelukast in other studies. There is no study in the literature evaluating the association of montelukast and levocetirizine combination therapy with the development of neuropsychiatric findings. Therefore, studies are needed to evaluate the neuropsychiatric effects of levocetirizine and montelukast combination therapy.

In our study, the most common neuropsychiatric finding developed after treatment was irritable behavior (13.2%). Nightmares (7.4%) were the second most common finding. Behavioral and mood disorder (5.9%), sleep disorder (4.4%), hallucination (4.4%), and restlessness (2.9%) were other neuropsychiatric findings observed after treatment. In a similar study conducted in our country, the most common neuropsychiatric effects related to montelukast in children were reported to be behavioral disorders, nightmares, and sleep disorders, similar to our study [23]. In another similar study in the literature, the most frequently observed neuropsychiatric effects after montelukast treatment were reported as irritability, aggressive behavior, and sleep disorders [24]. In a different study conducted in children in the literature, sleep problems and agitation were observed most frequently after the use of montelukast, similar to our study [35]. According to the literature and our study, the most common neuropsychiatric effects seen with the use of montelukast in children are similar. Children should be evaluated in terms of common neuropsychiatric findings such as sleep disorders and irritable behaviors in the combined treatments of montelukast and levocetirizine.

In this study, improvements in neuropsychiatric findings were seen. The percentage of patients with improvement in at least one neuropsychiatric finding was 76.5%. Improvements were mostly observed in sleep disorder, irritable behavior, and nightmares. All findings showed a decrease in percentage after treatment compared to before. This decrease was statistically significant, excluding hallucinations. Neuropsychiatric findings can also be seen in asthma and allergic rhinitis patients due to clinical findings caused by the disease itself, independent of drug use. For example, nasal congestion can frequently cause sleep disorders in patients with allergic rhinitis [36]. Similarly, sleep disorders due to symptoms such as cough can be seen frequently in asthma patients [30]. The neuropsychiatric findings that were present before the combination therapy of montelukast and levocetirizine was started at the beginning of our study may be due to the current clinical findings of asthma and allergic rhinitis. The decrease in the frequency of all neuropsychiatric symptoms after the treatment may be due to the regression in the clinical findings of the patients thanks to the treatment. The improvement after treatment observed in TRACK and RCSS scores in our study supports this assumption.

When the factors that may be related to the neuropsychiatric findings developing after the combination therapy of montelukast and levocetirizine were examined, no significant relationship was found between the development of neuropsychiatric findings in any of the factors such as gender, age, TRACK score, RCSS score, and allergy test positivity. In a study in the literature, similar to our study, no correlation was found between the development of montelukast-related neuropsychiatric effects and age and asthma severity in children [34]. There are also studies in the literature reporting a relationship between the age of children and the types of neuropsychiatric effects. For example, it has been reported that depression and suicidal thoughts are more common in adolescence, and sleep disorders are seen at younger ages [25]. Further studies are needed to explain the underlying pathophysiological mechanisms related to the development of neuropsychiatric effects and the factors that predispose to the development of side effects.

### Limitations and Strengths

The limitations of our study are that our study was conducted in a single center with a small sample size. Another limitation is the evaluation of neuropsychiatric effects that develop only in the short term. More data on the development of neuropsychiatric effects may be obtained in studies with longer follow-up periods. To the best of our knowledge, the lack of studies in the literature evaluating the neuropsychiatric effect profile of montelukast and levocetirizine combination therapy in children makes our study strong due to its contribution to this field. In addition to the development of neuropsychiatric findings, our study, which provides data on the improvement of neuropsychiatric findings observed in children after treatment, adds a broad perspective to the literature in this field. In addition, disease control, which we evaluated with TRACK and RCSS scores in children, improved after treatment compared to the pre-treatment period. Thus, our study presented an idea about the efficacy of montelukast and levocetirizine combination therapy, as well as evaluating the development of neuropsychiatric effects after the montelukast and levocetirizine combination therapy had started.

## 5. Conclusions

The development of one or more neuropsychiatric findings was observed in 22.1% of the children after the combination therapy of montelukast and levocetirizine. In addition, three out of four children showed improvement in the neuropsychiatric findings that were present before the treatment. While the most common neuropsychiatric findings were irritable behavior, nightmares, behavior, and mood disorder, the improvements were mostly seen in sleep disorders, irritable behaviors, and nightmares. In further studies with larger samples, factors that may be associated with the development of neuropsychiatric findings should be evaluated.

## Figures and Tables

**Table 1 children-10-01301-t001:** Laboratory parameters of the patients.

	Median (Min–Max)
WBC * (10^3^/uL)	8480.0 (3330.0–21,440.0)
Neutrophil (10^3^/uL)	3450.0 (1080.0–14,500.0)
Eosinophil (absolute)(10^3^/uL)	260.0 (0–1760.0)
Eosinophil (%)	3.0 (0–16.9)
Lymphocyte (10^3^/uL)	3840.0 (1140.0–13,510.0)
Platelet (10^3^/uL)	334,000.0 (167,000.0–743,000.0)
Total IgE (IU/mL)	91.0 (2.0–2735.0)

* WBC: White blood cell.

**Table 2 children-10-01301-t002:** Allergy test positivity of patients.

Allergy Test Positivity	n	%
House dust mite	14	20.6
Egg	5	7.4
Cat	3	4.4
Nuts	3	4.4
Pollen	2	2.9
Cow’s milk	2	2.9

**Table 3 children-10-01301-t003:** Symptom scores for asthma and allergic rhinitis of the patients.

	Median	Minimum	Maximum	*p* Value
TRACK-Before	40.0	5.0	75.0	<0.001
TRACK-After	87.5	25.0	100.0	
RCSS-Before	2.17	0.83	3.0	<0.001
RCSS-After	0	0	3.0	

TRACK: scored between 0–100 points. A higher score indicates better disease control. A score of over 80 points was considered well-controlled asthma. RCSS: varies between 0–3 points. As the score increases, the severity of allergic-rhinitis-related symptoms increases.

**Table 4 children-10-01301-t004:** Neuropsychiatric findings in patients before and after the treatment.

Findings	Before		After		*p* Value
	n	%	n	%	
Irritable behavior	56	82.4	43	63.2	0.004
Hallucination	8	11.8	5	7.4	0.508
Headache	14	20.6	2	2.9	<0.001
Nightmares	35	51.5	17	25.0	<0.001
Sleep disorder	43	63.2	22	32.4	<0.001
Behavioral and mood disorders	25	36.8	15	22.1	0.013
Restlessness	31	45.6	16	23.5	<0.001
Depression	0	0	0	0	-

**Table 5 children-10-01301-t005:** Neuropsychiatric effects and improvements in neuropsychiatric findings in patients.

Findings	Development of Neuropsychiatric Findings		Improvement in Findings	
	n	%	n	%
Irritable behavior	9	13.2	25	36.8
Hallucination	3	4.4	7	10.3
Headache	0	0	12	17.6
Nightmares	5	7.4	24	35.3
Sleep disorder	3	4.4	31	45.6
Behavioral and mood disorders	4	5.9	18	26.5
Restlessness	2	2.9	19	27.9
Depression	0	0	0	0
At least one neuropsychiatric finding	15	22.1	52	76.5

Development of neuropsychiatric findings was determined according to the Likert-type responses performed after the treatment, at least one unit increase compared to the pre-treatment. Improvement in neuropsychiatric findings was determined according to the decrease of at least one unit of Likert-type responses performed after the treatment compared to the pre-treatment.

**Table 6 children-10-01301-t006:** Factors associated with the development of neuropsychiatric findings.

	*p* Value	OR	95% C.I. for OR	
			Lower	Upper
Gender	0.754	1.210	0.368	3.983
Age	0.966	1.013	0.558	1.839
TRACK-Before treatment	0.806	1.006	0.962	1.051
RCSS-Before treatment	0.514	0.706	0.248	2.008
Allergy test positivity	0.438	1.649	0.466	5.834

## Data Availability

Not applicable.

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
