# Peer review of "Evaluation of Neuropsychiatric Effects of Montelukast–Levocetirizine Combination Therapy in Children with Asthma and Allergic Rhinitis"

_children, 2023, doi:10.3390/children10081301_

Round 1

Reviewer 1 Report

General comments:

The paper, titled as "Evaluation of Neuropsychiatric Effects of Montelukast Levocetirizine Combination Therapy in Children with Asthma and Allergic Rhinitis", byUÄŸur AltaÅŸ, Zeynep Meva AltaÅŸ, Fırat Öz, to explore Neuropsychiatric Effects of Montelukast Levocetirizine Combination Therapy. This is a good study in allergy field. However, there are something that needed to clarify before drawing some conclusions.

Specific comments:

1. How long is the patient's treatment time, and is the same dose for children with asthma and rhinitis? This is not explained in the text.

2. How effective was the treatment of the disease? Are there any other concomitant medications? The results of this need to be supplemented.

3. Since two diseases are involved, asthma and rhinitis, can they be analyzed separately?

4. This study is a combination of two drugs, and which drug caused the neuropsychiatric effect? Relevant analysis is required? Relevant analyses are needed.

Manuscripts written in English are complete and appropriate.

Reviewer 2 Report

This is an interesting study on montelukast+levoceterizine in small children 2-5 and its side effects. This subject is of interest of clinicians as particularly montelukast is a debatable drug due to its reports of suicide incidents. Data and study is novel, interesting and definitely worth publishing. Key conclusion is that up to 20% of patients might develop some neuropsychiatric side effects. On the other hand neuropsychiatric gain that is due to better disease control is much higher. Also appearance of side effects cant be predicted so far.

Though data and science work are ok, the paper needs to be improved. Some of the passages are difficult to read, some redundant sentences were found. I recommend text editing by native speaker or other qualified personnel. Also I listed some issues below that need to be adressed.

Introduction:

Any rationale behind using levoceterizine and not desloratadine that is also registered in this age group?

Lines 120-123 These two sentences seem identical in meaning.

Lines 61-63: "Montelukast is also used in the treatment and prophylaxis of asthma [12]. Montelukast can also be used in the treatment of allergic rhinitis as an alternative or combination therapy to oral antihistamines or nasal corticosteroids [13]." Please rephrase; these sentences sound bad together. Also consider removing prophylaxis or change to "prophylaxis of asthma attacks"

Line 72: remove "however"

Methods:

Where was the study conducted?

Section 2.2.2. requires a reference.

Results:

"The median age was 3 years (1.0-6.0)" How is this possible if children 2-5 were enrolled?

"When the laboratory values of the patients were examined, the median WBC and neutrophil values were 8480.0 103/uL (1080.0-21440.0) and 3450.0 103/uL (128.0-14500.0), respectively." Please correct neutrophil range according to table it should be 1280-14500. Also how is this possible to have minimum neutrophils 1280 and minimum WBC 1080?

There needs to be clarification what allergy tests were used. specific serum IgE? Skin prick tests?

Line 193 " was significantly lower than" please change to "significantly increased". Same thing line 198; it is so difficult to read. Consider external text editing.

Discussion:

Line 250-254 I suggest removing paragraph. Redundant

Line 255 sentence about treatment usage of levoceterizine and montelucast requires reference.

Line 327 Please remove sentence "This is another strength of our study."

Conclusions:

"According to the results of our study, at least one neuropsychiatric finding devel- oped in approximately one in 5 children after the combination therapy of montelukast and levocetirizine." This sentence is extremely difficult to read

In my opinion entire conclusions section needs to be rephrased to contain at most 3 sentences with most important findings or take-home messages.

References:

4. Is there any newer epidemiological study? This one seems old.

Though data and science work are ok, the paper needs to be improved. Some of the passages are difficult to read, some redundant sentences were found. I recommend text editing by native speaker or other qualified personnel.

Round 2

Reviewer 2 Report

Thank you for great response. All issues have been aswered. All in all, great manuscript.